# Combining crosshole and reflection borehole-GPR for imaging controlled freezing in shallow aquifers

Peter Jung[1], Götz Hornbruch[2], Andreas Dahmke[2], Peter Dietrich[1], Ulrike Werban[1]

[1] Department Monitoring and Exploration Technologies, Helmholtz-Centre for Environmental Research, Leipzig, D-04318, Germany

[2]Institute of Geosciences, Kiel University, D-24118, Germany

*Correspondence to*: Ulrike Werban (ulrike.werban@ufz.de)

**Abstract.**

During test operation of a geological latent heat storage system as a potential option in the context of heat supply for heating and cooling demand, a part of a shallow quaternary glacial aquifer is frozen at the "TestUM" test site. In order to evaluate the current thermal state in the subsurface the dimension of the frozen volume has to be known. As the target is too deep for high resolution imaging from the surface, the use of borehole Ground-Penetrating-Radar (GPR) is being investigated. For imaging and monitoring of a vertical freeze-thaw boundary, crosshole zero-offset and reflection borehole-GPR measurements are applied. The freezing can be imaged in the zero-offset profiles (ZOP), but the determination of ice body size is ambiguous, because of lacking velocity information in the frozen sediment. Reflection borehole-GPR measurements are able to image the position of the freezing boundary with an accuracy, determined through repeated measurements of $\pm 0.1\ m$, relying on the velocity information from ZOP. We have found that the complementary use of ZOP and reflection measurements provides a fast and simple method, to image freezing in geological latent heat storage systems. Problematic is the presence of superimposed reflections from other observation wells and low signal-to-noise ratio. The use in multiple observation wells allows for an estimation of ice body size. A velocity model derived from multiple ZOP enabled to extrapolate geological information from direct-push based logging and sediment cores to a refined subsurface model.

## 1 Introduction

High proportion of space heating is still based on burning fossil fuels (Steinbach et al., 2020). For the transition to renewable efficient heating, alternative concepts have to be developed and improved. Additionally, the predicted climate warming will result in an increased need for cooling applications (Lozàn et al., 2019). Using heat pumps, latent heat storages (LHS) allow the extraction of high amounts of energy for heating with little temperature change and low storage volume (Agyenim et al., 2010). For phase change materials with a low freezing point, such as water, the residual frozen volume generated during the heating period can act as a cold storage and be later used for cooling purposes. Therefore, LHS can meet the demand for both, energy efficient heating and cooling. In conventional LHS, tanks containing the phase change material are burrowed, which

makes installation expensive and limits building opportunities, especially in densely occupied urban areas, where there is a high heating and cooling demand (Steinbach et al., 2020).

Accessing geological layers, specifically aquifers, as storage volume, instead of burying tanks, can increase application possibilities, because of less above-ground space required and reduce installation costs. With segmented heat exchanger probes, that allow depth orientated controlled freezing and thawing (Fig. 1), the geotechnical use of aquifers as LHS is possible (Dahmke and Schwarzfeld, 2022).

To validate and improve thermo-hydraulic modelling of geological LHS, the current thermal state at operating depth has to be assessed. Data from borehole in situ temperature measurements and cooling fluid feed and return temperature only give punctual information. Since the boundary between frozen and unfrozen ground indicates a major change in thermal energy, knowing the lateral extent of frozen ground around the heat probes can act as a proxy for the thermal state, so imaging the lateral propagation of freezing with geophysical methods is desirable.

Extensive experience in the use of geophysical methods for the observation of thawing and freezing processes in permafrost soils and glaciers can be referred to (e.g. Campbell et al., 2021; Schwamborn et al., 2002; Terry et al., 2020; Vonder Mühll et al., 2002; Weigand et al., 2020). Electrical resistivity tomography (ERT) has been regularly used for monitoring of geological storages(Hermans et al., 2014; Hermans et al., 2015; Lamert et al., 2012). Though being sensitive to the strong change in electric conductivity between frozen and unfrozen sediment the application is not suitable for a precise estimation of a vertical freeze-thaw boundary. The lateral resolution of surface measurements like ERT decreases with greater depths and the high conductive layer on top of the monitored groundwater aquifer further decreases resolution in target depth. Even with borehole ERT and sharp boundary conditions applied in the inversion, the measurements at an LHS site would be affected by the installed heat exchangers and connecting pipes and cables.

The high contrast in electric permittivity of ice ($\varepsilon \approx 3 - 4$) and water ($\varepsilon \approx 81$) makes Ground Penetrating Radar (GPR) very sensitive to the phase change, which is therefore widely used for permafrost monitoring (e.g. Cao et al., 2017; Du et al., 2020; Hinkel et al., 2001; Sokolov et al., 2020; Steelman et al., 2010; Stephani et al., 2014; Stevens et al., 2008; Wang and Shen, 2019). Using borehole measurements gives the advantage of getting source and receiver close and perpendicular to the imaging target. Successful application has been shown in (Kim et al., 2007) with mapping ice rings in an underground liquid gas storage. Downside of the used setup is the non-directionality of the used antennas, making determination of reflector position ambiguous.

So far, the use of GPR has been examined in monitoring freezing and thawing processes within natural geosystems. However, its potential application as an indispensable measurement method for the operational monitoring of geological LHS systems remains unexplored. We hypothesize that borehole GPR will bring additional value in terms of spatial information of the ice body and thus can be used for monitoring purposes.

Therefore, the aim of this study is to use borehole reflection and crosshole GPR to image an ice body in a shallow aquifer. First crosshole GPR is used to get an enhanced site characterization before installation of a geological LHS system and then a concept of borehole reflection and crosshole GPR measurements is tested to image the ice development during plant operation.

## 2 Methods

### 2.1 Experimental site

The experimental site "TestUM" is located on a former airfield near Wittstock/Dosse in the northeast of Germany. The near subsurface consists of unconsolidated quaternary glacial sediments, which is a representative scenario for many areas in the
North European Plain. The "TestUM" site was already intensively used for several injection experiments. These prior experiments investigated the effects of injecting heat (Heldt et al., 2021; Keller et al., 2021; Lüders et al., 2021), carbon dioxide (Peter et al., 2012; Lamert et al., 2012) and nitrogen (Heldt et al., 2021; Hu et al., 2023; Peter et al., 2012; Lamert et al., 2012; Keller et al., 2021; Keller et al., 2024; Löffler et al., 2022; Lüders et al., 2021) into the subsurface. The subsurface investigations for these projects in the area show a high spatial geological variability typical for glacial sedimentation. There are layers of
fine, medium and coarse grained sands, glacial till (boulder clay) characterized by a mixture of sediments including high clay content and gravel alternating in thickness and lateral extent. Therefore prior to establishing the experimental LHS site, a comprehensive site investigation was conducted, using an adaptive approach that integrated surface geophysical measurements, direct push-based investigations, and traditional coring techniques. This multifaceted methodology was employed to identify the most suitable area for the experiment at the site. In particular, surface electrical resistivity and
electromagnetically induction measurements provided a non-invasive option to analyse subsurface characteristics in terms of near surface anthropogenic remains (old foundations, remains of the former airfield). Complementing this, direct push-based investigations with the hydraulic profiling tool (HPT, Geoprobe, USA) enabled a detailed examination of the geological setting with the help of vertical high resolution data of hydraulic and electrical conductivity (Mccall and Christy, 2020; Mccall et al., 2014; Vienken et al., 2012; Dietrich et al., 2008). Additionally, traditional coring was employed to extract core samples,
facilitating a more in-depth analysis of the geological strata. An area was selected for the experiment, where coring indicated a sandy aquifer at a depth of 10-16m, covered above and below by clayey till.

For testing of a geological latent heat storage, a subsurface volume is frozen and thawed in a controlled manner (Fig 1a). 16 heat exchanger probes, 16m deep each, are installed in a 1 m x1 m grid (Fig. 1b). The probes are designed with two separate fluid circuits, the upper spanning from 3 m to 9 m and the lower from 10 m to 16 m depth. Operating the lower fluid circuit
with feeding temperature <0 °C and the upper circuit with temperature >0 °C allows for depth orientated controlled freezing in the sandy aquifer while keeping the overlying geological units unfrozen. 18 2-inch wells and 9 multilevel wells are installed to have direct access to the subsurface for GPR measurements, in situ temperature measurements, and probing for

hydrochemical and microbiological analysis. The water table varies between $z_{water} = 2\,m - 3\,m\;bgs$ throughout the year, thus the experiment is taking place in a fully saturated environment.

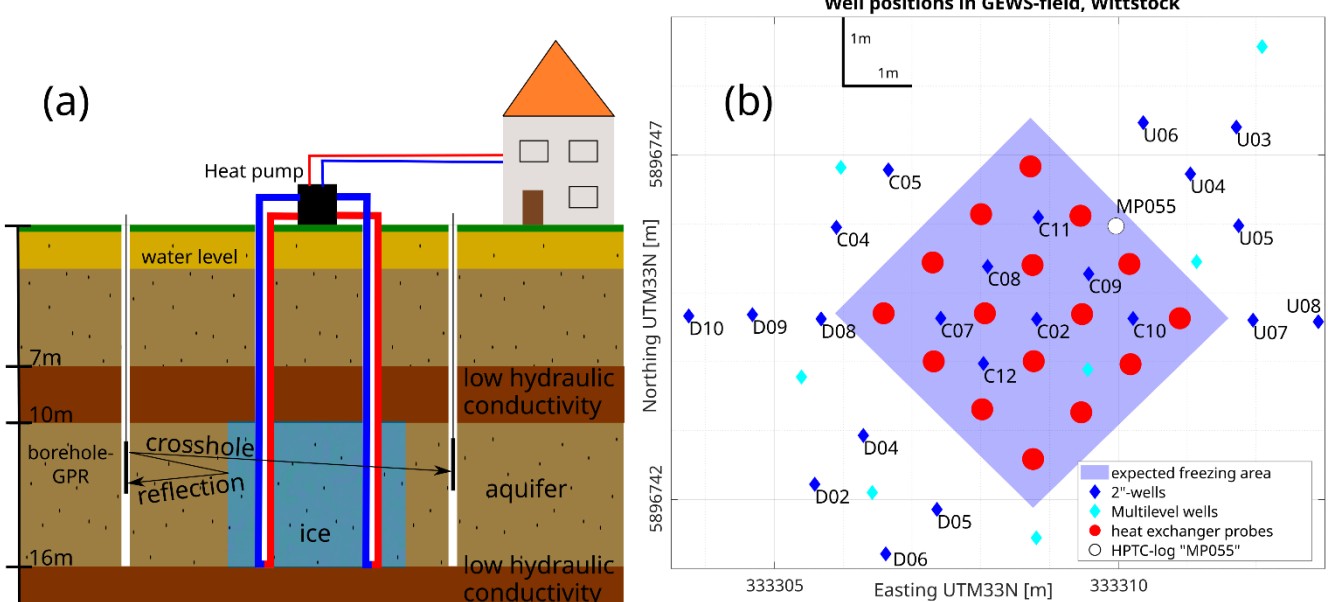

Figure 1: (a) Schematic sketch of geological latent heat storage at experimental site in Wittstock with concept of GPR-reflection and GPR-crosshole measurements. (b): Overview of the experimental site with position of borehole heat exchangers and observation wells.

The 2"-wells are 18 m deep with mesh filtering in 10 m to 16 m depth and therefore filled with water. Well naming is adapted to groundwater flow direction from north-east to south-west, which flows at a velocity of $0.05 - 0.09\,m\;per\;day$ (Heldt et al., 2021). Upstream wells begin with letter U, central wells with C and downstream wells with D. Wells for borehole GPR are positioned in 1.5 m distance from the outer heat exchangers to be close enough to register reflected signal despite the high attenuation caused by the high electrical conductivity in water saturated sediments. Closer wells would have a higher probability of being affected by freezing which would make them inaccessible. Also, if the reflector is too close to the antenna the direct wave superimposes the reflected signal. Wells are only on three sides of the experimental field, because an opening had to be left for logistical reasons, e.g. heat exchanger installation with heavy machinery.

## 2.2 Measurement methods

*Inclinometer*

Observation wells can deviate from their designated vertical orientation. The extent of deviation is measured using the DevProbe1 inclinometer (Geotomographie, Germany). Readings of tilt and heading are taken for every borehole, allowing calculation of the well path in the subsurface. Mean of occurring horizontal deviation in 17 m depth is 0.28 m with a maximum

of 0.99 m at well C05 due to a drilling obstacle. The corrected true well position in the subsurface is used for GPR interpretation.

*Ground Penetrating Radar*

The use of GPR reflections can give specific information about the distance from a well to a reflector, assuming a well-known propagation velocity. Therefore, the use of GPR is favoured in this setting. With our target being a vertical layer boundary and in depth greater than 10 m, surface GPR is not applicable, because the overlaying aquitard with high electrical conductivity

limits penetration. With the help of boreholes source and receiver antenna are brought close enough to the imaging target to record signals despite the high attenuation. Two omnidirectional Tubewave-100 (Radarteam, Sweden) borehole antennas, with vertical offset between centre of transmitting and receiving array of 0.3 m, in combination with a GSSI SIR-4000 (GSSI Geophysical Survey Systems, Inc., USA) are used for the survey. Peak frequency, measured inside the observation wells at the experimental site, is 190 MHz. We performed three measurements at three different stages of the test site operation: (1)

before installation of the heat exchangers for enhanced site characterization, (2) a baseline in the final equipped site and (3) a measurement after 54 days of running the LHS system when the existence of an ice body is verified by temperature measurements. For (1) we measured ZOP. To image the lateral extent of a body of frozen saturated sediment in (2) and (3), both, reflection measurements and ZOP are used.

| reflection profile | used v(z)-profile |
|---|---|
| C04 | C04-C12 |
| C05 | C05-C12 |
| D02 | D04-C08 |
| D04 | D04-C08 |
| D05 | D05-C10 |
| D06 | D05-C10 |
| D08 | C04-C12 |
| U03 | C09-U04 |
| U04 | C09-U04 |
| U05 | C10-U05 |
| U06 | C11-U06 |

Table 1: Velocity profiles from ZOP used for time-distance conversion of reflections.

For reflection measurements one single antenna acts as transmitter and receiver. It is lowered

down in a borehole and traces are collected every 0.25 m over an interval from 0.8 m to 17.3 m depth of midpoint between transmitter and receiver. Measurements are done in the wells D02, D04, D05, D06, D08, C04, C05, U03, U04, U05, U06. The remaining wells outside the freezing area are occupied with other instrumentation and the wells between the heat exchangers are inaccessible during freezing. A standard processing, consisting of subtraction

of DC-shift, zero-time correction, bandpass frequency filtering using 3$^{rd}$-order Butterworth-filter with cut-off frequencies 50 Hz and 600 Hz and a gain correcting for spherical divergence, is applied. The maximum phase of the reflection arrivals is picked. Maxima are picked, because due to low signal-to-noise-ratio first break picking is too erroneous. To account for time shift between first break and first maximum the traveltime of first maximum of the direct

wave is used for zero-time correction. Because emitted signal is assumed to be reflected at the freeze-thaw boundary and traveling back to the receiver (Fig. 1a), with known antenna position and propagation velocity, the distance to the appearing reflectors can be calculated. For distance calculation, borehole-specific velocity-depth profiles (see Table 1) are taken from the crosshole measurements described in the following paragraph.

For crosshole zero-offset profiles two antennas, one used as transmitter and one as receiver are placed in two boreholes. The antennas are lowered simultaneously measuring at the same depth every 0.25 m over a depth interval from 0.8 m to 17.3 m. The resulting ZOP are processed with the same processing flow as the reflection profiles. Zero-time is determined by picking the maximum of first arrivals in air measurements outside the wells. Source-receiver distance is increased from 1 m to 3 m in 0.2 m steps in the zero-time measurements. Subtracting the calculated traveltime for $v = c$ in air leaves the value for zero-time correction. From the first arrival maxima in the ZOP, we can calculate the bulk velocity of the soil between the antennas. When freezing happens we assume higher velocity and lower attenuation that will result in lower traveltimes and higher amplitudes. To get a 3D-distribution of wave velocity in the experimental field different well combinations are realized (Fig. 2a). Well combinations covering all three sides of the test field ensure velocity information in close vicinity to all reflection measurements.

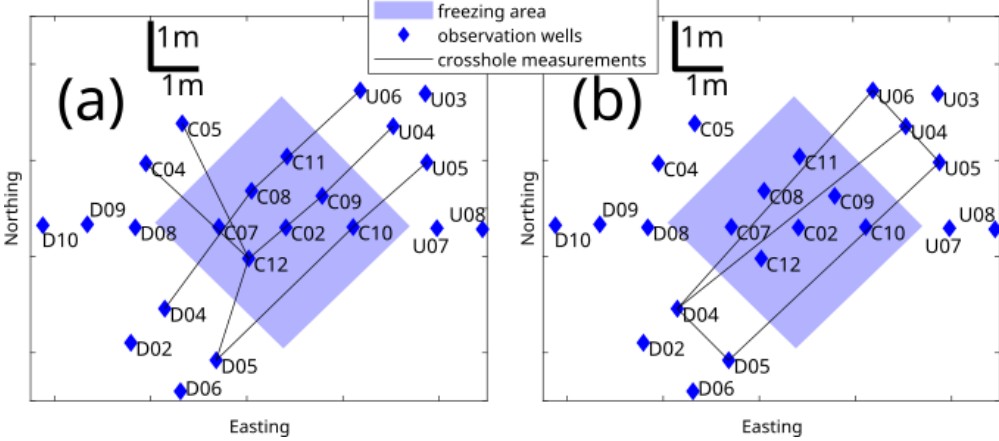

**Figure 2: Crosshole ZOP for (a) site characterization and (b) freezing monitoring.**

For the monitoring of freezing and thawing only measurements running through the freezing area are of interest (Fig. 3b). Measurements between wells outside the freezing area serve as reference to check for acquisition errors. If the measured velocity remains the same for the reference measurements, where there is no change in subsurface properties, the data acquisition is considered to be correct.

*Preliminary considerations - accuracy*

Accuracy of velocity estimation in ZOP is mainly influenced by positioning error and time error. Antenna positioning error consists of the GPS error and inclinometer error. Time error is mainly composed of picking error and zero-time drift of the aperture. Time error was assessed by comparing traveltimes of repeated measurements on multiple days of the same well combination. Figure 3a shows the expected velocity error assuming a positioning error of $\Delta s=0.05$ m and time error of $\Delta t=0.5$ ns. Greater distances between source and receiver minimize the velocity error. The downside of greater source-receiver distance is that lateral variations are averaged. Smaller distances ensure that the measured velocities, which are then used for time-depth conversion of reflection measurements, represent the immediate vicinity of the observation wells. So, for site

characterization a trade-of between high velocity error and spatial averaging is selected with well distances of 2 m to 4 m resulting in an expected maximum velocity error of $\Delta v < \pm 0.0025\ m\ ns^{-1}$ (Fig. 3a). ZOP for freezing monitoring, where the wave travels through the whole experimental field covering distances of around 6 m, reduces velocity error to $\Delta v \approx \pm 0.001\ m\ ns^{-1}$.

The low signal-to-noise ratio in reflection measurements results in an increased time error of $\Delta t = \pm 0.75\ ns$. Using the velocities from site characterization for determination of reflector position, velocity error and time error result in a distance error of $\Delta d = \pm 0.1\ m$ (Fig. 3b) at a reflector distance of 1.5 m. The 1.5 m correspond to the distance of the surrounding observation wells to the heat exchanger probes. As the freeze-thaw boundary moves outwards, and therefore the distance to the observation well decreases, the error decreases.

Vertical positioning was done aligning depth markings along the antenna cable with the top of the well casing. We assume the error of this method to be smaller than 0.01 m, therefore it was neglected.

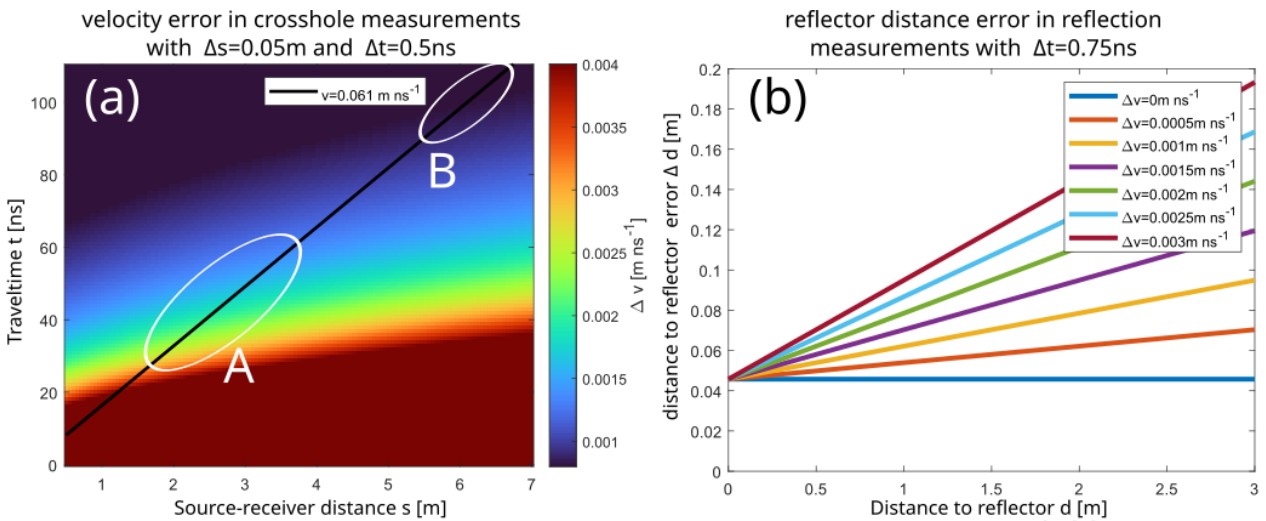

Figure 3: (a) Error of velocity estimation in crosshole measurements. Error depends on source-receiver distance and traveltime. The
black line indicates expected values for velocity of 0.061m/ns. White circles highlight ranges for A site characterization and B freezing monitoring. (b) Distance error for reflector position estimation via reflection measurements for different velocity errors.

The estimation of the lateral extend of the frozen body from ZOP velocities is assessed. The distance $s$ travelled by the wave is consisting of distance travelled in the thawed medium and in the frozen medium.

Traveltime $t$ is consisting of time travelled in the thawed medium and in the frozen medium:

$$s = s_{ice} + s_{thawed}, \quad t = t_{ice} + t_{thawed} \qquad (1)$$

Inserting in

$$t = \frac{s}{v} = \frac{s_{ice}}{v_{ice}} + \frac{s - s_{ice}}{v_{thawed}} \qquad (2)$$

and solving for $s_{ice}$ yields

$$s_{ice} = \frac{s * v_{ice}(v - v_{thawed})}{v(v_{ice} - v_{thawed})} \qquad (3)$$

If permittivity and therefore velocity of frozen saturated soil is unknown, the equation is underdetermined. Distance $s$ is known from GPS positioning corrected for borehole deviation, $v_{thawed}$ is taken from baseline measurements and $v$ is measured velocity. There is no access to direct information on $v_{ice}$ in the test field, because wells within the freezing zone are

inaccessible during plant operation. Nonetheless we expect to get qualitative information about beginning of freezing and shape of the ice body by looking at the changes of bulk velocity.

## 3 Results

### 3.1 Site characterization

Logging with the direct push-based hydraulic profiling tool (HPT) at drilling MP055 (Fig. 4d) shows a layer with high relative hydraulic conductivity $K_{HPT} \approx 10 \, ml \, min^{-1} \, kPa^{-1}$ and electrical conductivity of $\sigma \approx 10 \, mS \, m^{-1}$ between 10 m and 17 m depth. Above a layer with lower hydraulic conductivity and higher electric conductivity is observed. The low conductivity spans up to 7 m depth, while electric conductivity is higher at 10 m - 9.5 m and at 7 m depth. Below 17 m electric conductivity increases to $\sigma = 15 - 20 \, mS \, m^{-1}$ and $K_{HPT}$ decreases to $\sim 0.5 \, ml \, min^{-1} \, kPa^{-1}$. Sediment coring at MP055 (Fig. 4e)

matches the high hydraulic and low electric conductivity with a sand layer and the low hydraulic and high electric conductivity with higher clay content. We assume a subsurface model with a sandy aquifer in 10 m to 17 m depth covered on top and bottom with an aquitard. The low conductivity zone is very heterogeneous at this location (see supplement of Löffler at al., 2022). This can also be seen in Fig. 4 b + c: over the small test field we see most variations in amplitude and velocity precisely between 7 m and 10 m, indicating the presence of a highly heterogeneous boulder clay. We correlate the low amplitude and

low travel time with a highly heterogeneous layer, acting as a low hydraulic conductivity layer due to the presence of clayey material.

The ZOPs measured before installation of the heat exchangers offer the possibility for an enhanced characterization of the experimental site itself. ZOPs were used to spatially extrapolate the information over the area of the test site. One example profile from well C07 to C10 is shown in Fig. 4a. In the first 3 m the first arrivals are superimposed by a signal refracted at the

surface. Up to a depth of 7 m high amplitudes and longer traveltimes, converting to velocities of $v = 0.06 - 0.065 \, m \, ns^{-1}$, can be seen. From ~7 m to ~10 m and from ~16.5 m on there is layers with lower amplitudes and shorter traveltimes converting to velocities of $v = 0.07 - 0.075 \, m \, ns^{-1}$. Between 10 m to 16.5 m very high amplitudes and longer traveltimes, converting

to velocities of $0.06 - 0.065 \ m \ ns^{-1}$, are measured. The layering corresponds to the identified layers in the core drilling and

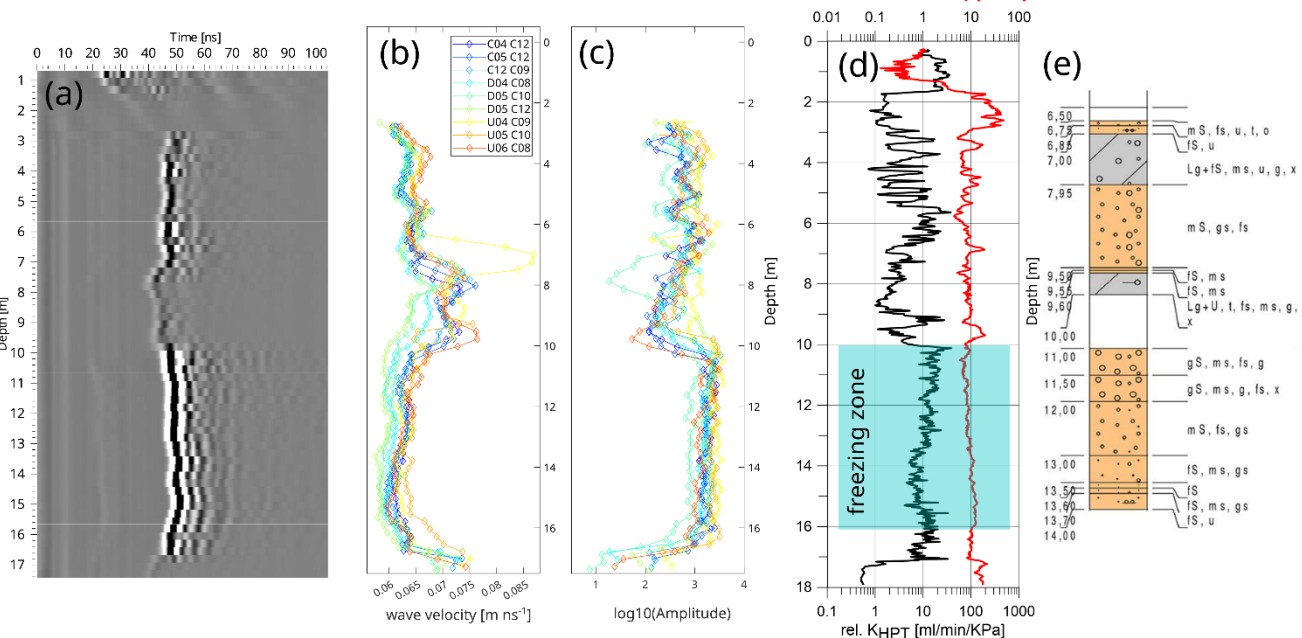

Figure 4: (a) Example profile from well C09 to C12. (b) Velocities calculated from first arrivals of all ZOP shown in Fig. 2a. (c) Amplitude of first arrivals. (d) Results of HPT- and EC-logging. (e) Sediment core at drilling MP055 (Position see Fig. 1b).

HPT- and EC-logging. The aquifer appears as layer with high amplitudes due to lower electrical conductivity and longer traveltimes due to higher permittivity, and the aquitard as layer with low amplitudes due to high electric conductivity and shorter traveltimes due to lower permittivity. Plotting all measured well combinations shows, that the layering of an aquifer covered on top and bottom with an aquitard is seen in velocity and amplitude in all profiles, but with thickness and velocity of the upper aquitard being variable. Placing the velocities at the midpoint between transmitter and receiver gives by the 2D view an idea about the 3D-distribution of the velocities over the experimental site (Fig. 5). Figure 5a is with a fixed north coordinate and Figure 5b with fixed east coordinate. A slight velocity increase is seen from south-west to north-east. Thickness of the upper aquitard also increases from south-west to north-east.

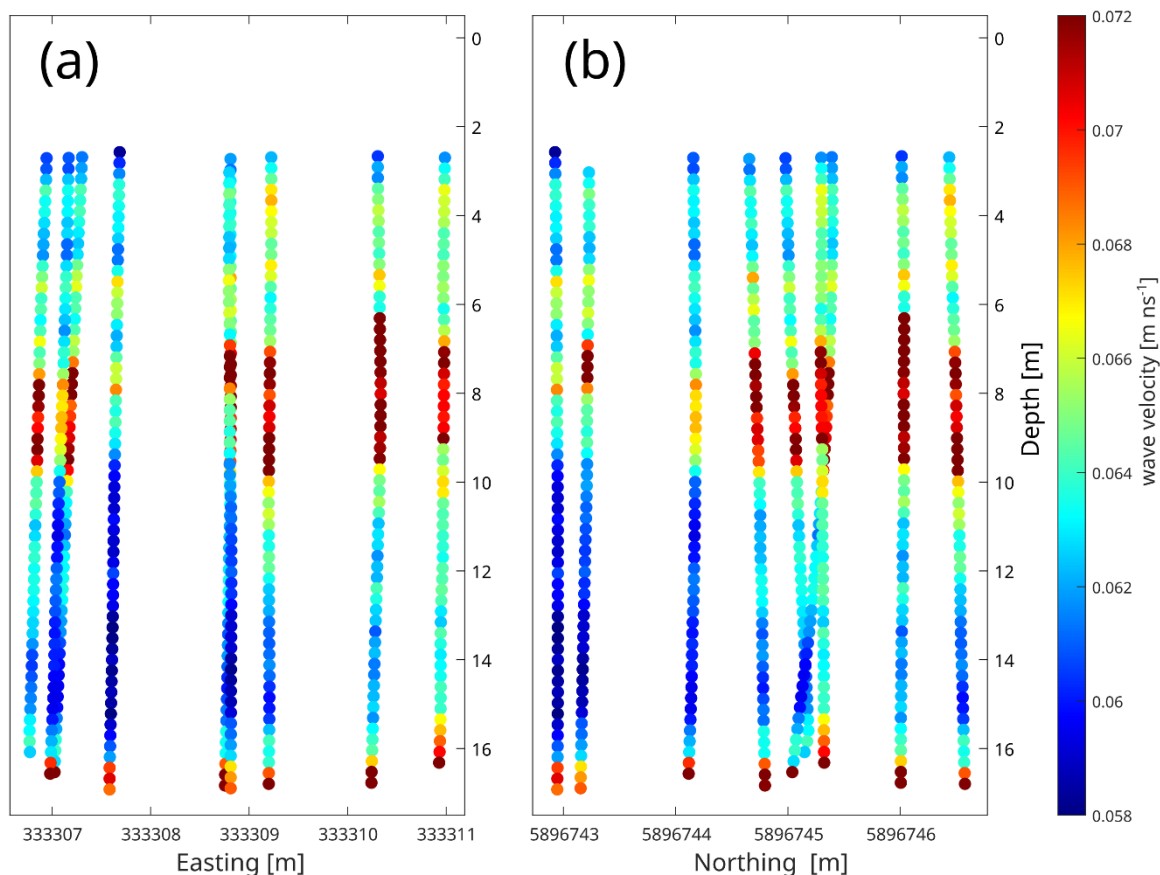

**Figure 5: Wave velocities placed at the midpoint between source and receiver. Lateral 2D-projection of 3D-distribution in (a) east and (b) north direction.3.2 Reflections profiles**

Reflection measurements in the finally equipped site were performed before the start of the freezing experiment and after 54 days of running the LHS system, with an ice body around the heat probes. In the baseline measurements linear reflectors are visible in reflection profiles. An example is shown in the profile from well D02 (Fig. 6a). In the lower aquifer reflectors are identifiable to a maximum traveltime of 80ns. In the aquitard, where attenuation is higher, only close reflectors are visible. From 0 m to 4 m depth the well reflections are superimposed by reflections at the surface and the water table. Picking

traveltimes of the reflectors and converting to distance, using the velocity estimation from ZOP, it corresponds to the distance

to other observation wells. So, in close vicinity reflections of the surrounding observation wells occur in the data. Not all surrounding wells can be identified because of low signal-to-noise ratio and superimposing signals.

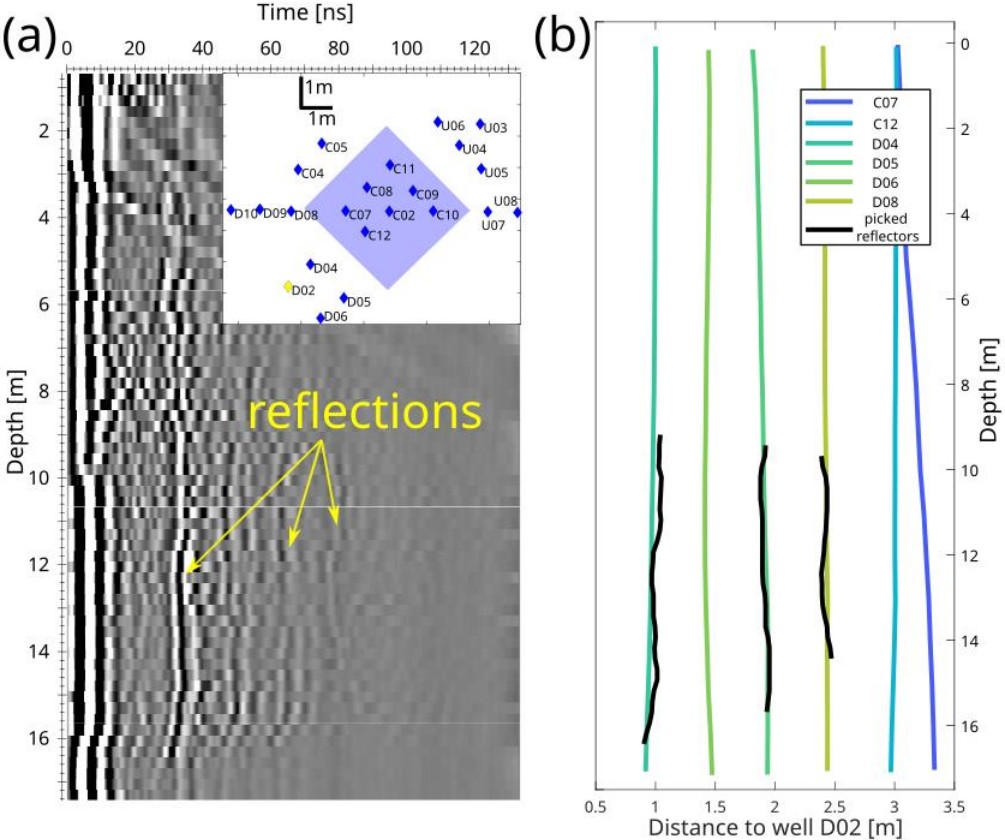

**Figure 6: (a) Reflection profile from baseline measurement at well D02 with visible reflectors. (b) Picked reflection traveltimes converted to distance using velocity from ZOP. Reflector positions coincide with distance to other observation wells.**

By identifying these reflectors as well reflections before the freezing is initiated, the baseline measurements enable to distinguish between well reflections and the signal reflected at the freeze-thaw boundary.

Figure 7 displays profiles from well U04 before freezing and with developed ice body during plant operation. A strong reflector appears at traveltime ~50 ns in depth of 10 m to 16 m. The hyperbolic increase of traveltime at the upper and lower end of the reflection indicate that the reflecting object ends there, rather than the signal is just not visible because of the high attenuation of the surrounding layers.

In a second step traveltimes of new occurring reflections are picked in all reflection profiles. After converting to distance the extent of the ice body is estimated. Because the antennas transmit the signal omnidirectional the possible reflector origin is a sphere around the antenna position. Assuming the reflector origin at the same depth as the antenna $\pm 0.2\,m$, the possible reflector positions of each measurement are plotted on a discretized 3D space. The positions closest to the test site are

interpreted as the freeze-thaw-boundary and connecting the edges gives an estimation of current lateral ice volume. Figure 8 shows the estimated ice boundary in a depth of 15 m with an extension of $4.3\ m \pm 0.2\ m$ in south-west to north-east direction.

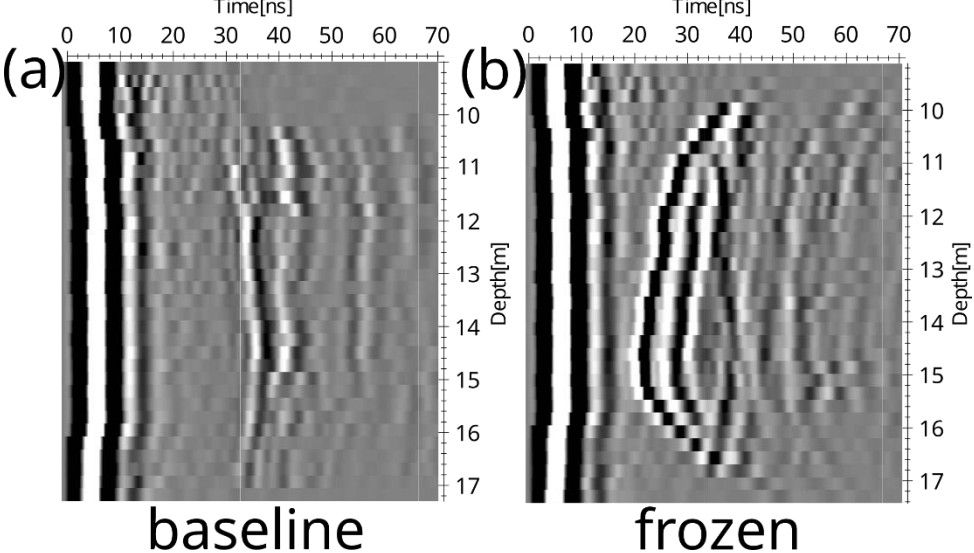

**Figure 7: Section of a reflection profile from (a) baseline measurement and (b) during freezing. Appearance of a new reflector in 10m-16m depth.**

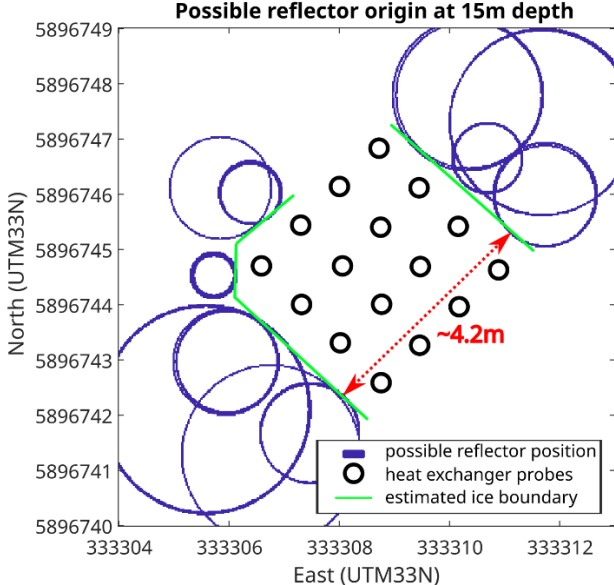

**Figure 8: Estimated ice boundary based on possible reflector origin.**

**3.3 Crosshole ZOP**

For the monitoring of freezing and thawing, only the well combinations crossing the whole freezing area D04-U04, D04-U06 and D05-U05 are of interest (Fig. 2b). Measurements perpendicular are not possible due to lack of observation wells in the south-east. The profiles D04-D05 and U06-U05 outside the freezing area serve as reference. They show unchanged velocities within the error range.

Figure 9 shows two profiles through the freezing area from well D04 to U04 and D05 to U05 before (Fig. 9a + c) and during

freezing (Fig. 9b + d). Due to greater distance between wells and the high attenuation, no signal is registered in the aquitard. With formation of ice the first arrival times in 10 m to 16 m depth drop significantly and reverberations occur. As an example, velocity in 15 m depth increases from $0.0605 \pm 0.001\ m\ ns^{-1}$ to $0.099 \pm 0.001\ m\ ns^{-1}$. Signal in depth greater than 16 m arriving with unchanged traveltime of ~100ns indicates no freezing there. The shown changes also appear in the third profile crossing the freezing area, which is not shown here.

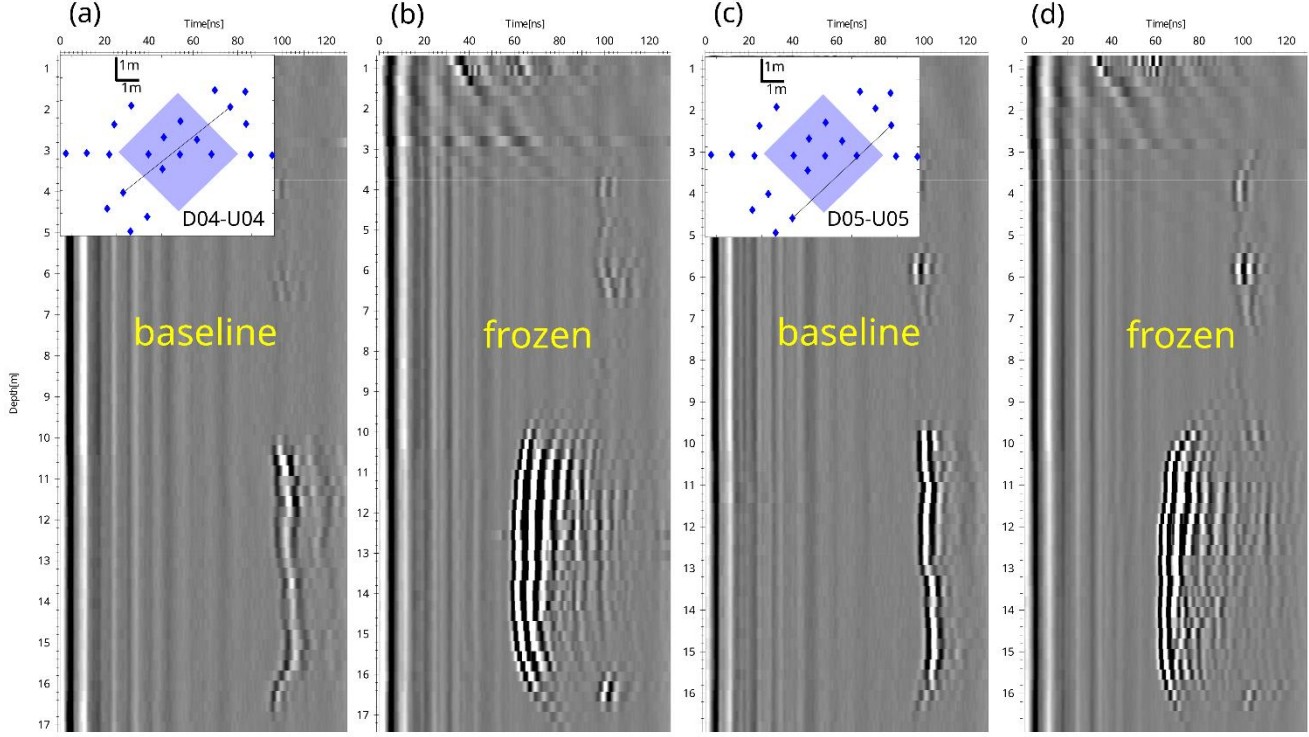

**Figure 9: ZOP between D04-U04 and D05-U05 before (a + c) and after (b + d) 54 days of subsurface freezing.**

**Discussion**

We are able to characterize the subsurface structure of the test site and image the frozen subsurface volume with crosshole and reflection borehole-GPR measurements. The accuracy with which the extent of the frozen ground can be estimated is affected by several sources of error. The methodology with its sources of error, limitations and underlying assumptions are discussed in the following subsections.

*Deviation correction*

The relevance of the well deviation measurements is highlighted by comparing the distances of the imaging target and the distance between the wells with the measured deviations. Mean deviation of ~0.3 m in 17 m depth can add up to a 0.6 m under or overestimation of distance between transmitter and receiver. For the longest distance, used in this study, of 6 m between well positions at the surface, this results in a 10% velocity error. For shorter distances the error increases. Therefore, deviation measurements are essential for borehole-GPR and recommended for all scenarios where well deviation is a possible source of error.

*Site characterization*

The crosshole ZOP measurements, which are required to obtain velocity information for the time-distance conversion of reflections measurements, were consistent with the drilling and logging data. Based on this correlation we were able to extrapolate this punctual information to the whole test site. This can be used for modification and verification of thermohydraulic modelling of the LHS experiment. It should be noted that the resolution of the model is somewhat limited because the velocity values are always a lateral average of the volume between source and receiver. Though being smaller than the expected velocity error (see 2.2 / Fig. 3a), the observed velocity increase from $v = 0.61\ m\ ns^{-1}$ to $0.63\ m\ ns^{-1}$ in south-west to north-east direction seems to be a real feature, because repeated measurements rule out measurement errors as source for the lateral velocity trend and variability. This spatial inhomogeneity of the subsurface prevents the use of longer baselines which would result in improvement of the velocity error.

*Reflection measurements*

Reflections of the formed frozen subsurface volume are visible and allow determination of the distance from reflector to the observation well. The ambiguity of reflector position due to omnidirectional wave emission can be removed by taking multiple measurements at different positions, as the possible reflections then overlap at the true reflector position. In cases of controlled subsurface freezing the origin is clearly determinable with the assumption, that there is only change in subsurface parameters around the heat exchanger probes. Closer lateral spacing of observations would be beneficial for a more accurate imaging of the freezing front, which is contradictory to the following issue.

An unpleasant finding was, that reflections of other observation wells are present in the data. We expected these reflections to be too weak to be measured, because of the small well diameter of 0.05 m compared to a wavelength of $\lambda \approx 0.3\ m$. Although smaller in amplitude these well reflections appear to be superimposed on the imaging target signal, making it difficult to accurately determine ice reflection traveltimes. The layout of observation wells being designed for not only GPR monitoring, but also geochemical and microbiological probing and in situ temperature monitoring is not ideal for the GPR measurements,

so monitoring designs for future projects should avoid having other observation wells in the same distance range as the imaging target.

As the geometry of the experimental setup is well known, 3D forward modelling of the wave propagation could help to eliminate reflections from installed structures. This would require further effort to account for the heterogeneity of the subsurface and the unknown electromagnetic parameters of heat exchangers and well casings. Since we are measuring from

outside of the heat exchanger array, the ice front is closer to the observation wells than the heat exchangers. We expect the reflections from the ice body to arrive earlier than the reflections from the heat exchangers. Therefore, we decided to not do forward 3D modelling, as it is also subordinate to the intent of developing an easy to implement monitoring tool for LHS.

*Crosshole measurements*

Resolution of ray-based tomography is around the size of the first order Fresnel-zone (Dessa and Pascal, 2003). With the parameters of the test site and a source-receiver distance of 6 m this corresponds to a maximum width of the first order Fresnel-zone of 0.7 m. With the sub wavelength resolution of using full waveform inversion a more detailed image could have been generated (Klotzsche et al., 2010). Our decision using simple crosshole sounding rather than tomography is based onthree main considerations. (1) Acquisition is fast, taking less than 10 min per profile and processing is fairly simple, making it

efficient for monitoring in future non-academic settings. (2) Resolving a boundary parallel to the acquisition plane with high accuracy is only possible with high angle ray paths. These measurements are not feasible, because of high attenuation in the measurement area. And (3) the complex 3D setting makes modelling more difficult. The presence of a large number of signal altering objects outside the acquisition plane, such as heat exchanger probes, observation wells and rocks is likely to create artefacts in a 2D tomographic model. For interpretation of first arrivals in ZOP out of plane effects can be ignored, because

they are higher in traveltime than the first arrivals. Multi offset gathers (MOG) with small offsets could increase resolution especially in the site characterization while avoiding the problem of too long raypaths in this high attenuation environment. For further projects an investigation with MOG before installation of the subsurface structure of the LHS plant is desirable, yet the highly increased acquisition time (Looms et al., 2018) and the more complex evaluation compared to ZOP makes it impracticable for fast and simple monitoring.

One interesting finding is, that when the subsurface is frozen reverberations occur that are not prominent in the unfrozen state. Possible explanations are refractions on not yet frozen parts inside the freezing area, or that the impedance contrast between surrounding material and the heat probes encased in concrete is greater in the frozen state, so that the signal is scattered on the heat probes.

The determination of ice body size from ZOP velocities is prevented by the missing value for velocity of the frozen ground.

Literature values for electric permittivity of frozen saturated soil span from $e_r = 3 - 6$ (Smith and King 1981), (Cassidy and Jol, 2009; Stevens et al., 2008; Cassidy, 2009), which converts to velocities $v_{ice} = 0.12 \, m \, ns^{-1} - 0.17 \, m \, ns^{-1}$. Using values from the crosshole measurements in 15 m depth: $s$=6.05 m, $v_{bulk} = 0.099 \, m \, ns^{-1}$, $v_{thawed} = 0.061 \, m \, ns^{-1}$, possible ice body size is $s_{ice} = 3.5 \, m - 4.5 \, m$. So even without accounting for measurement errors the uncertainty is bigger than with reflection measurements. In addition, this assumption is only true for a homogeneous velocity of the frozen area and no

permittivity change with varying ice temperature. Even if wave velocity of the frozen volume is known, no information about the lateral distribution of ice is possible. Due to groundwater flow we expect a difference in ice propagation between upstream measurements and downstream measurements. While reflection measurements should be able to image this asymmetry in freezing and thawing, ZOP does not provide information on lateral changes. A quantitative determination of ice body size from crosshole measurements is not possible due to insufficient determinability of permittivity of the frozen sediment, yet they are

sensitive to freezing between heat probes inside the test field, which cannot be imaged by reflection measurements. Also, the obtained velocity information is imperative for reflection interpretation.

The pros and cons of both types of GPR measurements suggest combining both to complement each other.

*Uncertainty*

While positioning error is inherent to the error ranges of GPS and inclinometer, time error is assessed with repeated measurements with the same acquisition geometry as suggested by (Yu et al., 2020). Repeating the same crosshole measurements shows an average error of $\Delta t = \pm 0.5 ns$. Zero-time corrections of up to 1.5 ns had to be applied, emphasizing the importance of determining zero-time before acquisition, even when using the same equipment and setup. It was measured at the beginning and end of each field campaign, but can vary between single profiles and even within the same profile (Axtell

et al., 2016) . A significant change in zero-time occurred, when there was a high difference between outside temperature and subsurface temperature. Comparing the zero-time measurements with air temperature data showed a temperature dependency, which is why we decided on using the zero-time from measurements directly after the last borehole measurement, because then, antennas and cables were at the same temperature as in the borehole. For further measurements bringing the equipment to subsurface temperature by leaving it in the borehole before zero-time determination can increase accuracy of zero-time

correction. Cross correlation between traces of different acquisition dates, in a depth range unaffected by freezing, could yield an additional correction factor. Unfortunately, in the profiles for freeze monitoring, attenuation is too high above the freezing depth. Precision would benefit from an improved t0-correction.

In reflection measurements the low signal-to-noise ratio increased the time error to $\Delta t = \pm 0.75 \, ns$. The resulting error for absolute reflector position is determined to be $\pm 0.1 \, m$. Considering relative changes of the freezing front over time, the

positioning and velocity error can be neglected, because they are the same for all repeated measurements. This increases accuracy for temporal change to $\Delta d = \pm 0.05 \, m$.

A gradual transition zone from unfrozen to frozen medium alters the reflected signal and shifts the arrival time. In our considerations the transition is presumed to be a sharp boundary. This assumption is based on freezing experiments with sediment from the test site. The observed transition zone is under $0.01 m$ thick, making our assumption a good approximation.

During freezing and thawing cycles of the LHS plant, temperature in the boreholes is affected by the extraction and insertion of thermal energy. In the observation wells downstream the groundwater flow, the temperature can vary between 4°C and 8°C. A permittivity change of the surrounding subsurface due to varying temperature is not considered in the estimations, but might add to the uncertainty.

*Vertical freezing boundary*

In this study we concentrated on imaging a lateral boundary. Nonetheless indications for top and bottom of the frozen area are present in the data. In ZOP the unchanged signal in depth greater than 16m indicates the end of freezing there. In reflection data we see one side of a hyperbola at the top and bottom of the reflector, characteristical for a sudden vertical change of impedance. Even though spatial resolution seems to be limited by the trace spacing of 0.25 m, finer depth increments are not

reasonable, because of the measured peak frequency $f = 190\ MHz$ resulting in a Fresnel-zone width at a 6 m distance of

$$b_{max} = \frac{\sqrt{\frac{v}{f}*d}}{2} = 0.7\ m, \text{respectively } b_{max} = \frac{\sqrt{\frac{v}{f}*d}}{2} = 0.35\ m \text{ for reflection measurements at reflector distance of 1.5 m, making}$$

clear vertical separation impossible. We tried migrating the profiles, but we did not get favourable results, because our trace spacing of 0.25 m between each trace is too coarse. In the unmigrated profiles vertical extension of the frozen area might be overestimated. The main goal was to get the lateral extent of the ice body, which does not suffer from the lack of migration.


**Conclusion**

We conducted borehole-GPR measurements in a shallow quaternary glacial aquifer before and during operation of an experimental geological latent heat storage. Prior to this study it was difficult to predict the geological conditions in the immediate vicinity of the experimental site. We show that it is possible to extrapolate punctual geological information from

one drilling point using a 3D-velocity model derived from simple zero-offset crosshole measurements. This allows to include detailed geometries of the geological layering into thermo-hydraulic modelling approaches.

Furthermore, while operating the LHS system, we aimed to investigate the feasibility of imaging a vertical freeze-thaw boundary using borehole GPR in the, for GPR, challenging context of water saturated glacial sediments. We therefore carried out reflection and crosshole borehole measurements during an LHS storage experiment, in which a depth-horizontal subsurface

volume is frozen. These experiments confirmed that both, borehole crosshole and borehole reflection GPR, enable to image the frozen subsurface volume. However, only reflection measurements are able to quantify ice body size by determining the position of the freeze-thaw boundary with an error of $\pm 0.1\ m$. Measuring at multiple campaigns has shown fast acquisition

and good repeatability of the data. Resolution is mainly limited by timing error of the wave arrivals caused by low signal to noise ratio, because of high attenuation in a water saturated environment. This prevents the use of higher frequency sources for reflection imaging. At least two boreholes are required to obtain accurate velocity information and to map the extend of freezing in one direction. There is evidence for the vertical confinement of the ice body, but clear determination of top and bottom are limited by Fresnel-zone width.

For further projects observation well positions that are in same distance range as the expected freeze-thaw boundary have to be avoided. In this study the monitoring design was affected by having too many observation wells in a similar distance as the imaging target, due to requirements of providing access for not only geophysical monitoring.

Taken together, these results suggest that borehole GPR is a viable method for monitoring LHS systems. The combination of ZOP and reflection measurements are a suitable setup for quick imaging of the lateral boundary of a freezing subsurface volume.

**Data access**

Inclinometer measurements and borehole-GPR data are published in the PANGEA repository:

Jung, Peter; Pohle, Marco; Werban, Ulrike: *Borehole-GPR crosshole and reflection data from monitoring of freeze-thaw cycles in a geological latent heat storage system [dataset].* Helmholtz Centre for Environmental Research - UFZ, PANGAEA, https://doi.pangaea.de/10.1594/PANGAEA.971978

**Competing interests**

One Co-Author is a member of the editorial board of Solid Earth.

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
