# Peer review of "Combining crosshole and reflection borehole-GPR for imaging controlled freezing in shallow aquifers"

_EGUsphere, 2024_

## Referee Comment (RC2)

[referee-annotated manuscript omitted]

---

## Referee Comment (RC3)

[referee-annotated manuscript omitted]

---

## Author Comment (AC1)

Thank you for the comments and suggestions. We appreciate your effort and tried to make the remarked sections more clear and added relevant information. Please find our details answers below:

Line 17: "Geological latent". Unclear. Please, revise the statement.

-> Latent heat storage is a fixed term. The use of geological formations as storage volume is tested in this research project, therefore geological latent heat storage. This is explained in line 25-35.

Lines 22-60: Any link between your research and Equivalent Porous Medium models to enhance geothermal energy usage in quaternary deposits? See relevant literature below: - Case studies of geothermal system response to perturbations in groundwater flow and thermal regimes. Groundwater, 61, 255-273.

-> Of course the plant operation will affect groundwater flow and the thermal regime in the subsurface, however in this manuscript we do not address this topic. The focus of our work is on the feasibility of imaging the development of a frozen volume in the subsurface with geophysical methods.

Lines 43-44: "Electrical Resistivity Tomography (ERT) has been regularly used for monitoring of geological storages". Please, insert recent review paper that discusses electrical geophysical methods in the field of geothermal energy; Review of Discrete Fracture Network Characterization for Geothermal Energy Extraction. Frontiers in Earth Science, 11, 1328397:

-> Many thanks for the reference to the inspiring paper. We checked it carefully and do not see relevance of the proposed article, since it does not address the use of ERT and focuses on fracture characterization, while this project is set in unconsolidated sediments, where no fractures are present. The remark, that the whole project setting is in unconsolidated sediments will be added in the site description.

Line 60: You should have only one aim and 3-4 specific objectives looking at the content of this paper. Please, revise the final part of your introduction.:

-> The paragraph will be changed to specify the goals more clearly.

Line 63/64: Insert detail on the thickness of the quaternary glacial sediments and nature of the bedrock below.

-> We have no information on the total thickness of quaternary glacial sediments and only operate in the first <20m. Prior site investigations give no indications on being close to bottom of the quaternary deposit, so bedrock does not have an influence on our measurements.

Line 67: Provide detailed description on the sedimentary heterogeneities and lithologies in glacial deposits. Several experts in geophysics, hydrogeology and civil engineering might be not familiar.:

-> Many thanks for pointing out these issues. We will extend information concerning glacial deposits.

Line 181-190: Have you got information on the stratigraphy of the boreholes to describe the nature of low and high hydraulically conductive layers?:

-> We do have a core from drilling point MP055. See lines 186-188: "Sediment coring at MP055 matches the high hydraulic and low electric conductivity with a sand layer and the low hydraulic and high electric conductivity with higher clay content." Information will be provided in Figure 4 additionally.

Line 196: You should have cores to address the comment above.

-> See comment above, information on the stratigraphy are provided.

Line 340: Remind to the reader that the sediments are of glacial origin in the conclusions.:
- > We will add in the conclusion that our investigations are performed in glacial sediments.

Figure 1: The spatial scale of the figure is unclear.:

-> Thank you for pointing that out. Depths information will be added to get an idea about spatial scale.

Figure 6: The black line is unclear in the graph.:
-> The legend will be updated to clarify the meaning of the black lines.

---

## Author Comment (AC2)

Thank you for the comments and suggestions. We appreciate your effort and tried to make the remarked sections more clear and added relevant information. Please find our detailed answers below:

Section 2.1: Geological description of the area is poor. It does not say much if you state that there are "quaternary glacial sediments". More detailed description must be added. If this is glacial till.. what is the clay contet for example? As sediment properties will influence effectiveness of GPR as well as changes in volume of the soil if freezing will accour.

-> Many thanks for pointing out these issues. We will extend information concerning glacial deposits and soil properties.

Line 106: ERT resolution is affected by many factors (electrode spacing, applied electrode configuration, 2D or 3D survey, overall electrical properties of the soil and numerous settings that you can change during data inversion process) and it is not good idea to state that 10m level is some magical depth where resolution drops. Also it is much more better to have high conductivity layers then low conductivity layers for ERT. My suggestion is to avoid discussion about ERT at all. No necessity to include it in this paper. & Line 110: Also for GPR it is not streightforfard to get the distance. It is also product of your interpretation.

-> It was not the intention to sound like ERT resolution would drop significantly below a specific depth. We are sorry if this may have been formulated inadequately. Of course ERT resolution depends on multiple factors, yet resolving the exact position of a vertical layer boundary in the target depth of >10m is not very accurate, and further decreased by the high conductive layer above the target, because then less current runs through the aquifer. We will revise this paragraph.

RC Line 112: If you would previously described soil type in moire detail it would be clear why 10m is already far for GPR. In sandy sediments you can get in depth of 20+m with ~200 MHz.

-> The statement will be adapted to make clear, that determined position is indeed just an interpretation of GPR reflections converted to distance using measured velocity profiles. In combination with the improved site description we will make more apparent why GPR penetration is limited in our setting.

Line 128: What GPR signal propagation speed was used and how did you determined it? This is crytically important question that must be explained in detail. Yes later you explain method for ZOP, but whas obtained values used also for reflection data? Also I suggest to use not only velocity values but values of dielectric permittivity (neglecting all the complications and assuming that this equation is valid $v=c/\sqrt{e}$). Most GPR specialists use values of permittivty instead of velocity.

-> Thank you for pointing that out. This was indeed not evident in the manuscript. A table will be added to show which $v(z)$-profiles were used for time-distance-conversion and we will explain, why the profiles are used. We mainly use velocity instead of permittivity, because this is what we can directly derive using measured traveltime and knowing source-receiver distance. Also calculating distances to reflectors is the main target, which can be calculated using the velocity. Permittivity values will be added to make it more convenient for readers.

Line 175: Are there no spatial variations in GPR propagation speed without frozen zone? This must be explaned.
-> Yes, there are spatial variations as seen in the baseline measurements, which add to the uncertainty. Thanks for pointing that out.  We will address this in the section.

Line 175: I did not found explanation what value for Vfrozen was used?
-> No value for v_frozen was used. We can add a possible value range from literature.

Line 182: Obtained information regarding sediments indicate multiple layer situation. As a result explanation of GPR signal propagation speed determination in the section of methods must be improved in order to explain that different propagation speeds were used for different layers.
-> This will be clarified by the changes as suggested in your comment to line 128. We use the v(z)-profiles from baseline measurements as velocities in the different layers.

Line 287. Theoretically also relfection GPR resolution depends of FIrst Fresnel zone as we approximate area of the reflection via First fresnel zone.
-> Yes, reflection GPR resolution also depends on Fresnel zone width. We will change the paragraph to address this.

---

## Author Comment (AC3)

We are very grateful for your detailed review. All the formal corrections and recommendations for citation, rephrasing for better readability, consistency with formatting of formulas and the grammatical errors will be considered. Your further comments will be addressed as followed:

*RC2 Line 58: "maybe borehole and crosshole?"* – We will formulate this more precisely.

*RC2 Figure 1: "How good are the chances that you see the heat exchanger probes in the data? Dimensions? You could do some easy modeling in3D for example with gprmax once with and once without the exchanger. Can be homogeneous layers to keep the calculation times low."*

Depth information will be added to the sketch to give an idea about the spatial scale. Unexpectedly, the reflection data from before and after the installation of the heat exchanger show no observable change in signal at the travel times corresponding to the distance to the heat exchangers. So we assume, that the observed reflections are reflections of the observation wells. As the heat exchangers are located in similar distance to the boreholes as the other observation wells, it is possible that any signal that does occur is superimposed.

We did not model the heat exchangers, because we expect the reflection of the ice body to arrive earlier than the heat exchangers' reflections, because we are measuring from the outside of the heat exchanger array. The ice front is closer to the observation wells, than the heat exchangers. Additionally, we do not have information on the electromagnetic properties of the heat exchangers and their casing.

*RC2 Line 92: "What about temp changes during freezing and thawing in boreholes and possible effect on permittivity?"*

Temperature in observation wells are indeed affected by temperature changes ( 4°C-8°C). Changes are small compared to the changes through freezing, but we will assess this in the discussion.

*RC2 Line114: "In air or in medium?"*

In the medium measured in the aquifer. Thanks for pointing out that this has to be clarified.

*RC2 Line119: "Add offset between Tx and Rx"*

Information on receiver offset will be added.

*RC2 Line 122: "Did use a gain during the measurement and if you did you calculate this out?"*

We have always used the same gain during acquisition, but have not calculated it out, as we focus mainly on the travel time for structural information. Amplitudes are only compared between profiles of the same measurement at different times.

*RC2 Line 125: "not clear."* – We will modify the phrase to clarify the meaning.

*RC2 Line 132: "Did you do this before every ZOP or once per day? Mention"*

We did zero-time measurements once before the first ZOP and after the last ZOP. Time between these is about 2 hours. This will be added to the manuscript.

*RC2 Figure 2: "How large could be out of plane effects for exampel ZOPs D4-U4 or U6-D4?? I think in Haruzi et al. (2022) are some modellings to this topic. Please dicuss this also in the dicusssion a bot more."*

Thank you for recommending this interesting paper. We will expand the discussion on this. There is multiple heat exchangers and observation wells within the Fresnel-zone. Since we only consider the first arrivals we expect the out of plane effects to be higher in traveltime and not affect our assessment.

*RC2 Line 141: "Why no MOG and inversion? Indicate the time aspects and for first investigations not suitable."*

The layers with high attenuation make MOG unfeasible, because we get no signal when the receiver and/or transmitter are in the high attenuation depths. Even in ZOP the signal is very weak in the low attenuation layers. Additionally, we aimed for a fast monitoring strategy and MOG and inversion would need more acquisition time. We will include this in the manuscript.

*RC2 Line 155: "Add something about wavelength and Fresnel volume"*

We will add values and have a discussion about the width of the Fresnel zone..

*RC2 Line 177: "What about gradients of thawing and freezing at the boundaries? add in discussion"*

Many thanks for that question, which we also discussed already. Indeed, a gradual freezing/thawing boundary would alter the signal. Laboratory tests with sediment from the test site indicate a very sharp freezing boundary with a transition zone <0.01m (not yet published). Assuming this in the test site, the effect is neglectable.

*RC2 Line 184: "Do you also have porosity information from the HPT?"*

No, unfortunately we do not have porosity values from the test site.

*RC2 Line 184 and 194: "Where? In logging data not clear why between 7-10 m depth amplitude damped so much.... ", "I dont see this confirmend in the logging, or?"*

As written, the EC-log only show thin layers with higher electric conductivity are observed, while the decreased hydraulic conductivity is observed between ~7m-10m. In a few cases we find that the EC probe is not properly coupled and the electrical conductivities are lower than expected - both methods, HPT and EC, are therefore used in a complementary manner. To ensure proper sealing of the aquiclude after installation of 2" monitoring wells, we took a sediment core at MP055 (not shown in the manuscript because used only for on-site decision) between 6,50m and 14 m. The core shows loamy material from 7m-7.95m, medium sand from 7.95m-9.60m and loamy material from 9.50m - 10m. The core confirms the HPT results and shows that the EC log has some issues resolving the loamy material between 7 m and 7,95m. In any case, it is important to note that the low conductivity zone (aquiclude) is very heterogeneous at this location, see Supplement of Löffler at al., 2022. This can also be seen in Fig 4 B & C: over the small test field we see most variations in amplitude and velocity precisely between 7m – 10m, indicating the presence of a highly heterogeneous boulder clay. We will rephrase this to make a clearer distinction. We correlate the low amplitude and low

travel time with a highly heterogeneous layer, acting as a low hydraulic conductivity layer due to the presence of clayey material.

[Figure]

*Core at drilling point MP055;*
*Stable Hydrogen Isotope Fractionation of Hydrogen in a Field Injection Experiment: Simulation of a Gaseous H2 Leakage,*
*Michaela Löffler, Merle Schrader, Klas Lüders, Ulrike Werban, Götz Hornbruch, Andreas Dahmke, Carsten Vogt, and Hans H. Richnow*
*ACS Earth and Space Chemistry 2022 6 (3), 631-641*
*DOI: 10.1021/acsearthspacechem.1c00254*

RC2 Line 184: "Which one?"

Hydraulic conductivity. This will be corrected to improve readability.

*RC2 Figure 4: "Or are these reduced amplitudes only visible is this ZOP? Related to heterogenous of the subsurface and/or a lense with higher electrical conductivity?"*

The reduced amplitudes are visible in all ZOPs at the same depth as the increased velocity. This indicates that we are dealing with a whole layer of different material rather than a lens and/or subsurface heterogeneity.

*RC2 Figure 5: "I am not sure if I totally understand the figure. So actually, its a 3D "image" of all ZOPs in the center of the planes and then you rotate them in two directions? Maybe add some explanations in the caption or the "3D" image too? "*

We have formulated it improperly so far in the manuscript. We will extend the description to make clear, that the figure is a 2D view of a 3D distribution. Figure 5a is with fixed north coordinate and Figure 5b with fixed east coordinate. Due to poor visibility in the 3D view, we have not included the 3D image.

*RC2 Figure 6: "Why do not all traces start at "zero" time? is this related to the casing or the time-zero correction?"*

For the zero-time correction we picked the first arrival of the direct wave of every trace and used it for correction. The differences at the beginning of the traces seem to be related to the sediment surrounding the well at the corresponding depth.

*RC2 Line 224: "Could this also indicate gradients in the freezing?"*

Yes, it could also be due to the ice being further away from the observation well, but as the decrease is hyperbolic we interpret this as diffraction at the upper end of the ice body.

*RC2 Figure 9: "Why not showing the other two ZOPs? I like seeing data."*

We decided to use only this example, because the results of the other ZOP are similar and we did not want to put to many figures in the manuscript. We will test adding an image of the other data and re-evaluate, whether showing more data emphasises the findings.

*RC2 Line 291: "Maybe a combination possible. Looms et al. " Mapping sand layers in clayey till using crosshole ground-penetrating radar" combines both ZOP and MOG measurements. Like ZOP for mapping and MOGs for higher resolution information. Side note: Maybe in future, 3D forward models could include such structures."*

We agree that a combination would have increased resolution and provided additional information. Since our target was mainly the feasibility of imaging the lateral position of the freezing boundary, we decided on using only ZOP and reflection measurements. In further studies the comparison to data with MOG would be desirable. We will add information in the discussion.

---

## Author Comment (AC4)

*Dear authors,*

*I have reviewed your manuscript and find that at the present state, it is unsuitable for publication at this journal. Firstly, I find that the overall presentation of the manuscript is weak and there are several spelling and grammatical errors, poor quality figures and the content is not well structured. I would highly suggest that you forward the text to a native speaker or make appropriate corrections before resubmitting.*

*Regarding the methodology and results, I find that your approach is a rather standard application of GPR that is better suited for an applied journal. You do not propose a novel methodology, only an application of existing approaches. Furthermore, you do not assess the need of migration for your GPR data which makes you over/underestimate the dimensions of the frozen soil body. Although you perform a depth conversion (from traveltimes) you do not mention the need of migration, which would correct for the strong hyperbolic patterns observed in your measurements.*

*In your discussion you mention a 3D geological model which is unfortunately never presented in the manuscript. Your error analysis focuses on positioning errors of the boreholes themselves, but you never mention positioning errors of the user (i.e., when running the borehole antennas in and out of the borehole). In several occassions you do not adequately explain your approach (e.g., when correcting for first arrival picking).*

*Overall, I find that this manuscript needs a major revision before being accepted to any journal, and I find that EGU solid Earth is perhaps not the most suitable journal because your work is primarily applied. You will find detailed comments of my revision in the appended pdf file, present as comments (larger tasks) or highlights (smaller edits).*

*Regards,*

Thank you for your thorough review and constructive feedback on our manuscript. We appreciate the time and effort you have invested in evaluating our work.

We acknowledge the need for significant improvements in the presentation of our manuscript. We will address the spelling and grammatical errors and improve the quality of the figures. We will also seek the assistance of a native speaker to ensure the language is polished and clear.

Regarding the methodology, while we understand your point about the application of Ground Penetrating Radar (GPR) being considered standard, we believe our focus on process observation and structural exploration in a highly unusual experimental context brings a unique perspective that fits within the multidisciplinary scope of EGU Solid Earth. Our approach, though based on existing methods, is tailored to the specific challenges and conditions of our study, which we believe adds value to the current body of knowledge.

We also recognize the importance of discussing the need for migration in GPR data. However, due to the large trace distance used in our study, migration is not feasible or appropriate. The large trace distance results from our aim to develop a fast applicable measuring setup, which necessitates a compromise between resolution and efficiency. This lower resolution diminishes the effectiveness of migration in correcting for hyperbolic patterns. Instead, we have focused on depth conversion from traveltimes and have provided a robust analysis within these constraints. We will clarify this point in our revised manuscript to ensure it is well understood.

Regarding the 3D geological model, we will rephrase the manuscript and clarify our intention. We create a 3D velocity model shown in Figure 5. With the information from the logging one can extrapolate the geological layers according to the velocity model.

We will clarify our approach, especially in areas such as first arrival picking corrections, to ensure that our methodology is comprehensively explained and transparent.

While we understand your concern about the journal's suitability, we believe that EGU Solid Earth, with its focus on experimental and multidisciplinary research, is an appropriate venue for our work. Our study emphasizes structural exploration aiming for process observation in unconsolidated shallow aquifers, e.g. glacial aquifers, which align with the journal's scope.

We appreciate your detailed comments and will address each one carefully in our revision. Thank you again for your valuable feedback.

**major comments**

*l 124    are these monopole antennas? If so, how do you perform time zero correction?*

Each antenna contains source and receiver. Distance between centre of source and receiver will be added in the description. Zero-time correction is done by picking the maximum of the first arrival and shifting the traces by this time difference.

*l 127    Can you please elaborate on this correction? It is not clear from the text what you mean, at least to me.*

This will be rephrased to make it more understandable. Zero-time correction is described in the answer to the previous comment. The reason or using the maximum of the first phase instead of the first break is the low signal-to-noise ratio of the ice body reflections. Picking first breaks of the reflections is often ambiguous, while the maximum is more reliable.

*Fig 2    why are there no axis labels or units on these figures? Please include.*

Axis labels will be added. Units are contained in the scale bar within the figure.

*Fig 3    This figure can be improved. Some comments*

*Use Capitalization for figures*

*eg. Traveltime [m] and not traveltime t[m]*

*The vertical label on plot b is misspelled*

Thank you for pointing that out. We will correct the errors and make the capitalization consistent in all figures.

*l 147    I get (and mathematica too)*

*s_ice = (v + v_th*v_ice - s*v_ice)/( v_th - v_ice)*

*which does not reduce to your equation*

We are glad that you pointed that out. The last formula (3) is correct, but there is a mistake one step earlier in line 171 (equation (2). The beginning of the line has to be t=s/v=…. If this is the case, the steps are correct.

In  t=s/v  we substitute t=t_ice+t_thawed, so we get t_ice+t_thawed=s/v.

Now substitute t_ice=s_ice/v_ice and t_thawed=s_thawed/v_thawed.

We get s/v= s_ice/v_ice +s_thawed_v_thawed.

Substituting s_thawed=s-s_ice and solving for s_ice yields the formula given in the original manuscript.

*Fig 5     It is difficult to assess this figure given that you overlap the scatter points and often hide information from points below.*

Yes, there is overlaps of data points, but, from our point of view, it only impedes the visibility of one ZOP in Figure 5a (at Easting =333308.8m). However, we have formulated it improperly so far in the manuscript. We will extend the description to make clear, that the figure is a 2D view of a 3D distribution. We choose to display the data in this way because it shows the 3D distribution of the velocity better than the 3D plots we created.

*Fig 6     This figure is unnecessarily large, and now you capitalize the labels in contrast to previous figures. You also do not mention in the caption anything about the other wells*

*(shown in (b).*

*Also (a) and (b) overlap with axis label/ticks*

We will remove the overlap. The large size was chosen for better visibility. We will check if reducing the size will still give insight on all relevant details. We could not identify reflections corresponding to the other wells. We will point this out in either the description or the corresponding paragraph.

*Fig 7     Th3 hyperbolic patterns observed in this image are very typical in single hole GPR reflection data. Without migration, these hyperbolas will largely overestimate the size of your ice body. I suggest you perform migration with a suitable velocity model, even if v_ice should be inferred, to avoid ambiguous results. Or at least test igration approaches and comment on these.*

You are right, the hyperbolic patterns are diffractions from the upper boundary of the ice body. We tried migrating the profiles, but did not get favourable results, because our trace spacing is to coarse with 0.25m between each trace. We will include this is the discussion and mention, that vertical size will be overestimated. Our main goal was to get the lateral extent of the ice body, which does not suffer from the lack of migration.

*Fig 8     Why use parentheses here and in all other figures use square brackets? Units are missing (I assue meters)*

Thank you for pointing that out. We will correct the errors and make all figures consistent.

*l 255     Can you start your discussion with an overall assessment of your work? It is unclear to the reader what you plan to discuss. It seems like you are going directly into the possible sources of error, but an introductory paragraph should help.*

We will rephrase the beginning of the discussion, before focusing on the error sources.

*l 303     Have you tried migrating your data with these two velocities to see what are the resulting profiles? You are now creating a simple time to depth conversion which is inaccurate at the frozen state. So including (even) an erroneous s:ice velocity does not seem much different. Migration will at least allow you to avoid the stron hyperbolic patterns you get from scattering of the EM wave.*

We tried migration with the velocities derived from ZOP (see answer to Comment on Figure 7). The time-depth correction is still valid in the frozen state, because our reflected signal only travels in the unfrozen medium and we know the velocity distribution in the unfrozen medium from ZOP.

*l 326     Not true, you can still have positioning errors which are related to the depth positioning of your antennas in the borehole. You do not address these in your manuscript, but they can easily be there and in the order of cm usually*

You are right, there might also be vertical positioning errors. We did the vertical positioning with fixed markings along the antenna cable and aligned these with the top of the observation well, so we are confident that our vertical error is <0.01m. We will mention this is in the discussion.

*l 345     you do not present a 3D subsurface geological model in your manuscript.*

This will be rephrased. We create a 3D velocity model shown in Figure 5.  With the information from the logging one can extrapolate the geological layers according to the velocity model.

**minor comments**

*l 46 isn't this true for all geophysical inversion?*

*Can you not apply sharp boundary inversion for ERT?*

*The correct arguent would be potential field methods vs. wave propagaton methods, but the current argument on interpreting inversion results is ambiguous. Please be more precise on the actual gap.*

Thank you for this suggestion. We will be more precise on why a wave based method yields higher accuracy for the position of a vertical layer boundary. Geophysical inversion is always depending on interpretation, which is why we try to avoid it by deducting information from reflections.

*l 65     Can you elaborate on these experiments if they are relevant to the study, instead of listing 8 references? The reader is unlikely to go through all these. If they are not all relevant, it is sufficient to list a few.*

Good point, we will either reduce the references or mention the other experiments (heat injection, methane injection, $H_2$-injection).

*l 142     How can correct data acquisition be ensured if velocity stays the same? Data acquisition is dependent on the operator, not on the changing subsurface.*

*Please reprase.*

The underlying assumption is that the subsurface remains unchanged in the reference measurements. Consequently, if there is no change in the data, it can be assumed that there is no significant error in the acquisition process. This will be further explained in the manuscript.

*l 152     what do you mean here exactly? Please explain better.*

Yes, this might be a bit vague and will be formulated more precisely.

*l 199-201    How exactly do you make this inference from your results? Can you guide the reader?*

The result is inferred from the results in Figure 4. The paragraph will be rephrased to explain better how this is concluded.

*l 201-202    Figure 5 does not give a 3D impression of your experiment. A 3D figure would give such an impression.*

In our opinion, a plot with fixed Easting and fixed Northing coordinates can give an impression of the 3D velocity distribution. We tried 3D plots, but the 2D plots were clearer.

*l 205 / Fig 4    you call these profiles and radargrams interchangeably. Please be consistent.*

We will make the naming consistent.

*l 264    Where is this 3D geological model? Why is it mentioned here but never presented in the manuscript?*

See answer to comment on Line 345.

*l 279    Since you know the geometry so well could you try to remove these with forward modeling, e.g. gprmax 3D ? It also seems like the phase is different between these reflectors and the frozen section, so this could lead to some more effective processing.*

It would be nice to model this, but we lack information on permittivity of observation wells and heat exchangers. Moreover unexpectedly, the reflection data from before and after the installation of the heat exchanger show no observable change in signal at the travel times corresponding to the distance to the heat exchangers. So we assume, that the observed reflections are reflections of the observation wells. As the heat exchangers are located in similar distance to the boreholes as the other observation wells, it is possible that any signal that does occur is superimposed. Another observation is that not all of the observation boreholes actually appear as reflections - combined with geological heterogeneities, forward modelling was not used in this first experiment. Therefore, in a recent second experiment, we have avoided installing observation wells and heat exchangers at similar distances, which may allow us to test such a modelling approach. Results are expected in mid-2025.

*l 293    be consistent with writing 3D. The correct and used approach is to use 3D. you sometimes use 3 dimensional, sometimes 3D and here you spell out three-dimensional.*

We will make the naming consistent.

*l 319/320    Did you have consistent time zero at the beginning and end? Please include this information in the metohdology. You are only mentioning this here.*

We saw a change in zero time, when there was a high difference between outside temperature and subsurface temperature. Comparing with temperature data showed a temperature dependency, which is why we decided on using the zero time from measurements directly after the last borehole

measurement, because then, antennas and cables were still at the same temperature as in the borehole.

*l 321    you are mentioning this as time-zero up until now, when you mention t0. You can not expect the reader to understand a new and not-defined nomeclature at the discussion of your manusccript.*

We will make the naming consistent.

*l 325    computed or estimated? If computed, how?*

This will be changed to computed. For how, see Figure 3b.

*l 351    You seem to only do 3 campaigns, and one is only ZOP. So this would not be multiple (better write out 2 or 3)*

We did more test measurements to assess repeatability. This information will be added.

*l 354    this is only true if anisotropy is not present*

True. We consider no anisotropy in unconsolidated saturated sediments.